# Effects of Burosumab Treatment on Two Siblings with X-Linked Hypophosphatemia. Case Report and Literature Review

**DOI:** 10.3390/genes13081392

**Published:** 2022-08-04

**Authors:** Claudia Maria Jurca, Oana Iuhas, Kinga Kozma, Codruta Diana Petchesi, Dana Carmen Zaha, Marius Bembea, Sanziana Jurca, Corina Paul, Alexandru Daniel Jurca

**Affiliations:** 1Faculty of Medicine and Pharmacy, Department of Preclinical Disciplines, 1 December Sq., University of Oradea, 410081 Oradea, Romania; 2Regional Center of Medical Genetics Bihor, County Emergency Clinical Hospital Oradea, Romania, (Part of ERN THACA), 410469 Oradea, Romania; 3Department of Pediatrics, Victor Babes University of Medicine and Pharmacy, 300041 Timisoara, Romania

**Keywords:** X-linked hypophosphatemia, PHEX gene, FGF23, Burosumab

## Abstract

X-linked hypophosphatemia (XLH) or vitamin D-resistant rickets (MIM#307800), is a monogenic disorder with X-linked inheritance. It is caused by mutations present in the *Phosphate Regulating Endopeptidase Homolog X-Linked* (*PHEX*) gene responsible for the degradation of the bone-derived hormone fibroblast growth factor 23 (FGF23) into inactive fragments, but the entire mechanism is currently unclear. The inactivation of the gene prevents the degradation of FGF23, causing increased levels of FGF23, which leads to decreased tubular reabsorbtion of phosphorus. Clinical aspects are growth delay, limb deformities, bone pain, osteomalacia, dental anomalies, and enthesopathy. Laboratory evaluation shows hypophosphatemia, elevated alkaline phosphatase (ALP), and normal serum calcium levels, whereas parathormone (PTH) may be normal or increased and FGF23 greatly increased. Conventional treatment consists of administration of oral phosphate and calcitriol. Treatment with Burosumab, a monoclonal antibody that binds to FGF23, reducing its activity, was approved in 2018. Methods. We describe a case of two siblings, a girl and a boy, diagnosed with XLH, monitored by the Genetic Department of the County Emergency Clinical Hospital since 2019. The clinical picture is suggestive for XLH, both siblings exhibiting short stature, lower limb curvature, bone pain, marked walking weakness, and fatigue. Radiological aspects showed marked deformity of the lower limbs: genu varum in the girl, genu varum and valgum in the boy. Laboratory investigations showed hypophosphathemia, hyperphosphaturia, elevated ALP, normal PTH, and highly increased FGF23 in both. DNA analysis performed on the two siblings revealed a nonsense mutation in exone 5 of the PHEX gene: NM_000444.6(PHEX):c.565C > T (p.Gln189Ter). Results. At the age of 13½ on 7 June 2021, the two children started treatment with Burosumab in therapeutic doses and were monitored clinically and biochemically at regular intervals according to the protocol established by the Endocrinology Commission of the Romanian Health Ministry. Conclusions. The first results of the Burosumab treatment in the two siblings are extremely encouraging and suggest a favorable long-term evolution under this treatment.

## 1. Introduction

X-linked hypophosphatemia is a monogenic disorder and the most common form of hereditary rickets with a prevalence of 1:20,000–1:60,000 in newborns [1,2,3,4]. The clinical picture includes hypophosphataemia, disharmonic dwarfism, bone deformities (curvature of the lower limbs), osteoalgia, dental abscesses, osteomalacia, and enthesopathies. From a physiopathological point of view, phosphate metabolism is affected by mutations in the *PHEX* gene, which cause elevated serum levels of FGF23. The latter is involved in the tubular reabsorbtion of phosphorus and the production of calcitriol [5,6]. Elevated FGF23 levels cause loss of phosphorus in urine by suppressing the activity of the Na-phosphate cotransporter and by suppressing 25(OH)D3 1-hydroxylase expression in the kidney [7,8]. The mechanism by which mutations in *PHEX* gene increase FGF23 levels is still unclear, although both PHEX and FGF23 are derived from osteocytes [9]. The *PHEX* gene has a wide variety of mutations, with over 615 recorded to date [10,11,12,13,14]. Reported mutations are diverse and include frameshift mutations, missense, nonsense mutations, intronic splice-site mutations, and deletions; these are distributed throughout the gene with no hot spot regions. Conventional treatment for XLH consists of administering calcitriol and phosphorus, but this treatment has a wide range of side effects, such as nephrocalcinosis, hyperparathyroidism, and gastrointestinal disorders. Biological therapy with monoclonal antibodies (Burosumab) was introduced in 2018. Recently published phase 2 and 3 studies demonstrated the extremely promising therapeutic effects of Burosumab, including improvement in growth curve and bone pain, increased capacity for effort, and in biological terms a normalization of phosphorus, FGF23 and PTH levels [15,16].

The purpose of this article is to present the clinical evolution of two siblings diagnosed with XLH one year after initiating treatment with Burosumab.

X-linked hypophosphatemia rickets (XLH) is a severe and rare diseaseThe authors describe a familial case of XLHThe clinical picture is different for the two siblings, especially in terms of bone changesThe Burosumab treatment has favorable effects, improving growth curve and bone pain, and normalizing phosphorus levels

## 2. Materials and Methods

### 2.1. Case Report

We describe a familial case of two siblings referred to the Genetics Department of the County Emergency Clinical Hospital Oradea, Romania, for short stature and severe lower limb deformities. Case ≠ 1, a girl, was the first child in the family, coming from a pregnancy with physiological evolution, vaginal delivery at 40 weeks, birth weight 3000 g, and birth length 48 cm. Case ≠ 2, a boy, was the second child in the family, coming from a pregnancy with physiological evolution, vaginal delivery at 38 weeks, birth weight 4200 g, and birth length 50 cm. Family history: young, unrelated parents; the mother’s maternal grandfather had disharmonic dwarfism, excruciating bone pain, waddling gait, and moderate genu varum deformity of the lower limbs.

### 2.2. Laboratory Investigations

Laboratory tests were focused on the assessment of phosphorus, alkaline phosphatase, PTH and 1.25 (OH)_2_ vitamin D at periodic intervals of 3, 6 and 12 months, according to the protocol established by the Endocrinology Commission of the Romanian Health Ministry.

### 2.3. Molecular Investigations

Written informed consent was obtained from the mother prior to participation in the study. DNA was extracted from peripheral blood lymphocytes of the participating family members using standard extraction procedures. Molecular tests were performed at Synlab Laboratories, Lausanne. The patients were tested using next-generation sequencing (NGS) with a multi-gene panel of 13 genes selected based on associations with hypophosphatemia, as reported in the medical literature: *ALPL*, *CLCN5*, *CYP2R1*, *mCYP27B1*, *DMP1*, *ENPP1*, *FAH*, *FAM20C*, *FGF23*, *FGFR1*, *PHEX*, *SLC34A3*, *VDR*. Each gene in the NGS panel was targeted with oligonucleotide baits (Agilent Technologies, Santa Clara, CA, USA; Roche, Pleasanton, CA, USA; IDT, Coralville, IA, USA). Genomic DNA was extracted from the biological sample using the bead-based method. DNA quality and quantity were assessed using electrophoretic methods. The sequencing library was prepared by ligating sequencing adapters to both ends of DNA fragments and were size-selected using the bead-based method. Regions of interest (exons and intronic targets) were targeted using a hybridization-based target capture method. The sequencing libraries that passed quality control were sequenced by Illumina’s sequencing-by-synthesis method, using paired-end sequencing. Sequence alterations were confirmed using bi-directional Sanger sequencing. The variant classification follows the modified Blueprint Genetics Variant Classification Schemes from the ACMG guideline 2015. A heterozygous variant of PHEX was identified c.565C > T, p.(Gln189 *) and classified as pathogenic. Written parental consent was obtained in order to include case details and accompanying images.

## 3. Results

### 3.1. Morphological Evaluation of the Patients

The siblings, sister (case#1) and brother (case#2), were referred to the Genetics Department in 2019, at the ages of 10 and 4, respectively, by their family doctor (Table 1). Treatment with Burosumab was initiated in June 2021, and the phenotypic characteristics of the patients prior to starting the treatment are shown in Table 1.

Case#1. The girl, 13½ years old, displayed very short stature: 131 cm (−3 SD) and normal weight: 39 kg. Chest examination revealed pectus excavatum, thoracolumbar scoliosis, Harrison’s groove and rachitic rosary. Musculoskeletal examination showed thickened wrists and ankles, severe bilateral bowing of lower limbs (genu varum). She had walking difficulties: slow walking, waddling gait, and marked fatigue after covering short distances (50 m), which is why she made frequent stops; intercondylar distance was 39 cm. Ophthalmological exam showed amblyopia (Figure 1).

Case#2. The boy, 7 years old, also displayed short stature: 107 cm (−3 SD) and normal weight: 19 kg. He also presented craniofacial dysmorphism: macrocephaly, with an ocipitofrontal diameter of 53 cm (above the 90th percentile), flattened facies, frontal bossing, bilateral epicanthal folds, and high-arched palate. Chest examination showed pectus excavatum, Harrison’s groove and rachitic rosary. Musculoskeletal examination revealed thickened wrists and ankles, bilateral bowing of lower limbs and impaired walk (Figure 2).

### 3.2. Laboratory Investigations

Lab tests performed at the time of diagnosis and 1 year after the initiation of the Burosumab treatment are shown in Table 2. In the same table it can be seen that under the action of the Burosumab the values of phosphatemia and alkaline phosphatase changed in the direction of normalization.

### 3.3. Radiological Investigations

Radiological investigations show a genu varum deformity in both femural and tibial bones in Case#1. Case #2 presented genu varum deformity in femural bones; in tibial bones: left tibia genu varum, right tibia genu valgum (Figure 3 and Figure 4). These are summarized in Table 3.

### 3.4. Psychological Aspects: Case#1 Displays Low Self-Esteem Due to Motor Difficulties (Increased Emotional Sensitivity, Shyness in Unfamiliar Environments). Frequent Episodes of Sadness. Case#2 Periods of Anxiety and Anger

### 3.5. Molecular Investigations

NGS shows a pathogenic variant in *PHEX* gene c.565C > T, p.(Gln189 *). This variant generates a premature stop codon in exon and is predicted to lead to loss of normal protein function, either through protein truncation (188 out of 749 aa) or nonsense-mediated mRNA decay.

## 4. Discussion

The diagnosis of familial hypophosphatemia was based on clinical aspect and laboratory changes (hypophosphatemia, increased ALP, decreased 1,25-dihydroxy vitamin D, increased FGF23), as well as on the radiological aspect of the bones. The diagnosis was confirmed by molecular investigations which highlight the mutation in the *PHEX* gene.

### 4.1. The Gene

The *PHEX* gene is located on the short arm of chromosome X (Xp22.11) and was identified in 1995. It contains 22 exons and encodes a protein (PHEX protein) made up of 749 amino acids, belonging to the M13 protease family (type II cell surface zinc-dependent proteases) involved in the degradation of the extracellular matrix. The protein consists of an extracellular domain, where the active site is located (positions 581, 646), three zinc coordination sites, but also other glycosylation links and Cys Cys disulfide bonds. *PHEX* also has other functions not related to FGF23 degradation, which is expressed in osteoblasts and osteocytes, but also in other tissues (ovary, testicles, lung, brain, muscle). In vitro studies showed that *PHEX* cleaves an extracellular matrix protein called osteopontin that inhibits bone mineralization and is found in bones and teeth [17,18].

The *FGF23* gene is located on chromosome 12 and encodes a protein made up of 251 aminoacids. Although it is expressed by osteoblasts/osteocytes, its low levels are also expressed in other tissues in rodents (brain, teeth) [19].

The family of fibroblast growth factors (FGF) contains 22 de members involved in many metabolic processes such as angiogenesis, cell growth and differentiation, cell migration and tissue repair processes. FGF23 belongs to this family of proteins, and more specifically to the endocrine subfamily FGF19, along with FGF19 and FGF21 [20,21,22].

Although there is no direct link between *PHEX* and *FGF23*, the inactivation of the *PHEX* gene leads to increased FGF23 levels [23,24,25,26]. Phosphate and 1,25-dihydroxyvitamin D [1,25(OH)2D] levels are the main regulating factors of the *FGF23* expression. In circulation, once FGF23 is activated by osteoblasts and osteocytes, it binds to a transmembrane protein, more precisely to the α-Klotho cofactor, and modifies phosphocalcic homeostasis. Specifically, it stimulates phosphate excretion and inhibits the formation of 1,25(OH)_2_D3, the active form of vitamin D (1,25 dihydroxycholecalciferol) ([27]). By activating the PTH/PTH receptor in osteoblasts and osteocytes, PTH stimulates FGF23 synthesis (Figure 5). FGF23 decreases PTH synthesis and secretion (downregulation) in the parathyroid [28]. FGF23 bound to the FGFR1 receptor, in the presence of α-Klotho, inhibits the action of 1-α-hydroxylase enzyme (encoded by Cyp27b1) in the kidneys and FGF23 stops the synthesis of 1,25-dihydroxyvitamin D. On the other hand, PTH stimulates Cyp27b1 expression in the kidneys (upregulation) and increases the serum concentration of 1,25(OH)_2_D3 [29,30].

Besides phosphate and 1,25(OH)_2_D3, other regulatory factors such as calcium, leptin, estrogen, glucocorticoids, iron metabolism and bone mineralization are involved in FGF23 expression [31].

Elevated levels of FGF23 are found in the autosomal dominant form of hypophosphatemic rickets caused by activating mutations of FGF23c, and also in the autosomal recessive form (due to mutations in dentin 1 matrix protein).

Sarafrazi Sodabben et al. have developed a new PHEX gene locus-specific database, (PHEX LSDB) for the purpose of centralizing and disseminating information about *PHEX* gene variants. Data was collected from several databases as follows. 1: The older 2000 version of *PHEX* LSDB which is currently inactive. 2: Genetic data obtained from clinical trials of Burosumab. 3: Recent data published in the literature. 4: A state-of-the-art free molecular testing program initiated by Ultragenyx Pharmaceutical Inc. and Invitae Corporation [31,32,33]. By April 2021, 870 unique variants of PHEX had been identified and collected in PHEX LSDB from the four databases listed above; more than 800 disease-causing PHEX variants have also been found.

In our patients, the nonsense mutation was identified in exon 5 of the *PHEX* gene: c.565C > T (p.Gln189Ter). The PHEX c.565C > T, p.(Gln189 *) variant had been previously reported in four unrelated individuals affected with X-linked hypophosphatemia, of which two were reported to be sporadic cases (Table 4) [34,35,36].

### 4.2. Clinical Aspects

The disease is characterized by high morbidity and the appearance of bone deformities accompanied by severe bone pain, which affects the lives of patients [38]. These signs and symptoms affect the patients’ quality of life (QoL). In the case presented by the authors, the sister (case ≠ 1) who has reached the age of puberty, displays school adjustment issues, an inferiority complex, shyness, and reluctance in expressing her feelings.

The growth. Disharmonic dwarfism, with a predominant shortening of the lower body compared to the upper body, is the main feature in XLH, although it presents great variability [39,40]. The final stature in adults is −3 SD when compared to normal stature. The clinical picture is more severe in males than in females. The lower limbs are much more affected, with their marked curvature, because they are exposed to mechanical forces much more strongly than the upper limbs. It must be noted that growth plate activity is intense in the knee, causing the growth of the bone in length and the formation of the enchondral bone. Additionally, growth plate activity is often strongest around the knees, where it influences the rate of endochondral bone production and, consequently, bone length [41]. In adulthood, the clinical picture is dominated by the appearance of osteomalacia and enthesopathies associated with bone and joint pain due to secondary arthritis [42]. In our patients, Case#1 only registered an increase of 2.5 cm in height; at 1 year after the initiation of the treatment she measures 133.5 cm. Case #2 registered a 4 cm increase in height, and at 1 year after the initiation of the Burosumab treatment he measures 111 cm.

Dental abscesses. The appearance of dental abscesses is another main feature manifested from childhood, with devitalized teeth (especially mandibular incisor and canine teeth), moderate to severe periodontitis, or several asymptomatic periapical lesions on teeth that have already been treated endodontically or not can be seen in adults. The dental changes of patients appear as soon as the teeth erupt and persist throughout their lives, resulting in severe functional, aesthetic, and nutritional issues. The patients must be managed in specialized medical services, with constant attention paid to both functional and aesthetic rehabilitation. Our patients had not displayed any dental abscesses to date.

Craniofacial dysmorphism is discrete, and some patients may have cranial dysmorphism with macrocephaly and dolichocephaly [43]. Case #2 presents macrocrania, with a high forehead, frontal protuberances and flattened parietal bones.

### 4.3. Paraclinical Aspects

The major sign is the decrease of phosphatemia due to decreased renal tubular reabsorbtion. Alkaline phosphatase and FGF23 levels are highly elevated. Normal FGF23 levels do not rule out the presence of XLH, so in the presence of low hypophosphatemia, they should be interpreted as inadequately normal [44]. Increased levels of FGF23 are also present in other forms of hypophosphatemic rickets (autosomal dominant and recessive forms). PTH levels may be normal or at the upper limit, calcium level is low in most cases and 1,25(OH)_2_ vitamin D level is decreased. In addition to hypophosphatemia, our patients displayed elevated ALP and FGF23 levels, with normal levels of PTH.

### 4.4. Radiological Aspects

The changes that occur are those present in rickets, including enlarged, abnormally shaped metaphyses, and irregular diaphyses in the long bones; unlike in deficiency rickets, the cortical region of the bone is thickened, and bone resorption is absent. These changes are visible in areas of rapid growth, such as the distal part of the femoral and tibial bones or the distal part of the radius. An X-ray of the knee joint is sufficient to diagnose XLH, because the lower limbs are the first affected [45,46]. Our patients display characteristic changes of rickets. Case #1 displays genu varum deformation at femoral and tibial level and closed growth plates at femoral and tibial level. Case #2 displays femoral genu varum deformation, genu varum deformation in the left tibia, and genu valgum deformation in the right tibia.

### 4.5. Genotype-Phenotype Correlation

Although the disease manifests in all patients with *PHEX* gene mutations, severity may differ between members of the same family, irrespective of gender, as noted in the literature [47,48,49].

Mutations in the *PHEX* gene are extremely varied, including nonsense, missense, splicing, and frameshift mutations, so the pathogenic variants can be spread on the entire length of the gene [50,51,52]. There are currently more than 615 variants in this gene identified in the Human Genome Mutation Database (HGMD). According to Gaucher et al., the increased and diverse number of *PHEX* gene mutations makes the genotype-phenotype correlation less obvious [53]. Pathological variants lead to alteration of the protein, with alteration of protein function and truncated protein appearing in over 70% of the cases. This wide range of mutations disrupts the cell processing mechanism, alters the conformation of the protein, and also alters endopeptidase activity, as proven by numerous studies [54,55,56]. Some authors reported that women with certain variations in the PHEX gene have a milder clinical picture (moderately low levels of serum phosphate, milder bone deformations) than males with the same genetic variants. Other authors believe that pathogenic variants that lead to the formation of truncated proteins or variants in the C-terminal region of the gene increase the severity of the clinical and paraclinical picture [57,58].

Some authors, such as Cho et al., when comparing missense mutations and nonsense mutations, did not establish a genotype-phenotype correlation [59]. Although our patients have the same mutation, there are some differences: the bone deformities are more pronounced in the girl, with marked genu varum deformity and an intercondylar distance of 39 cm causing severe disharmonic dwarfism (with treatment, the intercondylar distance was reduced to 36 cm), whereas macrocephaly with a high forehead and frontal protuberances was present only in the boy.

### 4.6. Medical Treatment

There is currently no cure for XLH, so the goal of the current treatment is to improve bone deformities and other anomalies caused by severe hypophosphatemia [60].

The classical treatment of XLH consists of administering phosphate and calcitriol. Phosphate is administered in doses of 20–60 mg/kg/day, 4–6 times/day. The sooner the treatment is started, the more beneficial the effects are. A normalization of phosphorus and ALP levels is observed, and radiological alterations are improved; growth rate is also enhanced and bone pain is relieved. Calcitriol is associated with the phosphate treatment, in doses that vary from patient to patient. This treatment does not have the expected effects, both drugs having severe side effects, most commonly nephrocalcinosis, reported in 30–70% of patients, and secondary hyperparathyroidism, which further aggravates phosphaturia and bone resorption [61]. Our cases followed this treatment inconsistently for a year, after which they stopped it for unknown reasons.

Although growth hormone (GH) has beneficial effects on growth rate and PTH values (normalization of values), its administration is not recommended because its effects cause asymmetric growth in which some areas grow more than others, eventually leading to skeletal deformities and disproportion between body segments by stimulating chondrocyte proliferation, leading to increased growth rate without normalization of the growth plate in these patients [62,63,64].

Biological therapy with monoclonal antibodies that prevent the excessive activity of FGF23 is beneficial and is administered especially in patients with abnormal development of the growth plate. In 2018 Kaneko et al. demonstrated that blocking FGF23 in various ways leads to improvement in hypophosphatemia, bone integrity and also in enthesopathies [65,66].

Burosumab treatment for XLH was approved by the European Medicines Agency (EMA) in February 2018 in Europe, and in April 2018 by the US Food and Drug Administration (FDA) for patients older than 1 year. Treatment consists of subcutaneous injections with doses of 0.8 mg/kg/2 weeks, with the possibility of increasing the dose by 0.4 mg/kg if normal serum phosphorus levels are not reached, without exceeding the normal dose of 2 mg/kg. Burosumab bioavailability is almost 100% due to its good subcutaneous absorption [67,68]. The drug degrades into small molecules, and maximum levels are reached between 7.0–8.5 days, with a half-life of 16.4 days [69].

As a general rule, it is recommended that treatment with phosphate and calcitriol should be stopped one week before the start of treatment with Burosumab. The most common side effects are redness around the injection site, headache, and bone pain in the extremities.

Studies have shown that treatment over a period of 4 to 8 weeks increases femoral bone density, causes thinning in the growth plate, improves serum levels of phosphate and 1,25 (OH)_2_D3 [70]. If serum phosphorus levels do not reach normal levels after the first month of treatment, doses may be increased by 0.4 mg/kg to a maximum of 2 mg/kg. In our patients’ case, Burosumab doses were progressively increased until normal levels of phosphorus were reached: in the case of the girl the phosphorus values normalized after 3 months of treatment, while in the case of the brother after 4 months. With both patients a progressive increase of doses with 0.4 mg/kg was necessary. One year after the start of treatment, the doses administered were: 1.2 mg/kg/2 weeks (50 mg/2 weeks) for the girl and 1.4 mg/kg/2 weeks (40 mg/2 weeks) for the boy.

As it is a newly approved treatment, there are no studies to prove long-term efficiency and impact on the quality of life in adult patients undergoing this treatment [71]. It is certain that the Burosumab treatment improves the growth curve and anthropometric parameters without causing disharmonious growth, as it is the case with GH administration [72].

There are currently several uncontrolled open-label studies that have demonstrated the effectiveness of the Burosumab treatment. In an open-label phase 2 trial study, Carpenter et al. followed short term effects in 52 children, aged 1 to 12, diagnosed with XLH [73]. The aim of the study was to change the biochemical parameters to 40 and 64 weeks, respectively, after starting treatment, by following the Rickets Severity Score (RSS or Thacher score) [74]. RSS is a quantitative parameter measuring the severity of rickets according to radiological alterations in the wrists and knees. It measures the degree of wear in the mataphysis and the concavity and the percentage of growth plates damage. The results are interpreted on a scale from 1 to 10, 10 being the highest degree of severity, and 0 representing the absence of radiological changes specific to rickets. The authors of the study found that the effects of Burosumab treatment are beneficial: phosphorus levels reach the lower limit of reference values, and bone pain is decreased, while the capacity for effort is increased and children are able to walk longer distances until fatigue sets in.

In a randomized active controlled open-label phase 3 study (with data collected from 16 clinical sites), Imel et al. compared the effects of classical phosphate and calcitriol treatment and those of the Burosumab treatment in children with XLH. The study highlighted the favorable effects of Burosumab: a clinical improvement in rickets, in the growth curve and in biochemical parameters (phosphorus, PTH) [75,76]. In another phase 2 study, Whyte et al. showed the same beneficial effects of the treatment with this monoclonal antibody in children with XLH (Table 5) [77].

The effects of the Burosumab treatment in adults were highlighted in a recent double-blind placebo-controlled phase 3 study by Insogna et al. The effects of the Burosumab treatment were monitored over a 24-week follow-up period. A total of 134 adult patients with XLH were included in the study. Burosumab proved to be effective both in terms of biochemical parameters, specifically phosphorus (increasing serum phosphate levels) and in terms of osteoarthritis development [78]. Similar effects were highlighted by Cheong et al. in Japanese and Korean adults with XLH, in a multicenter open-level single-dose study [79].

### 4.7. Surgical and Orthopedic Treatment

Surgical treatment aims at obtaining an equal length in the lower limbs, their correct alignment at the end of bone growth, and maintaining a joint as mobile and functional as possible.

The basic surgery technique is osteotomy performed in places where bone deformity is major. The result is an immediate correction of the deformity by internal fixation or gradual correction by external fixation. This technique is not often used anymore, due to the fact that it has been associated with frequent complications and a high risk of recurrence. In 2017 Gizard et al. showed that up to 57% of patients surgically treated by osteotomy suffered from postoperative complications, such as recurrent deformities present in 29% of patients [80]. However, if the bone deformity is severe and causes instability in the knee, surgical treatment is recommended before the end of the growth process.

Nowadays surgical techniques are more efficient and less invasive. Guided growth surgery is becoming more and more frequent and is performed in early childhood. It is recommended if there is no improvement in bone deformity after 1 year of medical treatment. The purpose is to correct the growth plate defect of the deformed bone before marked diaphyseal deformity occurs. The principle is to correct the mechanical axis. The forces acting on the diaphysis and implicitly on the joint are brought back to normal, leading to normal growth, and the procedure does not require immobilization. The technique involves setting a metal plate on the lateral or medial surface of the bone (for genu varum or genu valgum deformities) at the level of the growth plate; the metal plate is fixed with screws placed one proximal and one distal. In time, bone alignment is achieved; when the desired result is obtained, the metal plates are extracted, bone growth following its physiological course. This technique helps correct the bone only in coronal plane, without improving the median correction of the tibial bone [81].

### 4.8. Diagnosis and Monitoring

Diagnosis and follow-up monitoring of XLH patients represent a real challenge for practitioners. The process involves a multidisciplinary team that includes a geneticist, a pediatrician, an orthopedist, an ENT specialist, a dentist, a rheumatologist, an internist, a nephrologist, an endocrinologist, a physiotherapist and a psychologist. In 2011, Carpenter et al. published the first XLH Guide, “Clinician’s guide to XLH” [82]. More recently, in 2019, Hafner et al. published an article on practical clinical recommendations for diagnosing, treating and monitoring XLH patients, providing important and precise details on all aspects of this condition from childhood to adulthood.

Genetic counselling is extremely important. Being an X-linked dominant disorder, the affected father will have all daughters affected, and the affected mother runs a 50% risk of having affected children, be they boys or girls. Molecular diagnosis, which identifies pathogenic variants, provides correct genetic advice and allows for the screening of relatives at risk [83].

## 5. Conclusions

X-linked hypophosphatemia has a wide spectrum of clinical presentations. Burosumab treatment has beneficial effects on the growth curve, on bone changes and also mitigates fatigue. At the same time, it improves the patients’ quality of life, as it is a drug that revitalizes the lives of XLH patients. In Romania, the number of undiagnosed cases is still high; currently only 15 patients are being treated with Burosumab. Genetic counselling plays a key role, while early diagnosis and interdisciplinary cooperation are essential for proper management leading to improvement of the patients’ quality of life.

## Figures and Tables

**Figure 1 genes-13-01392-f001:**
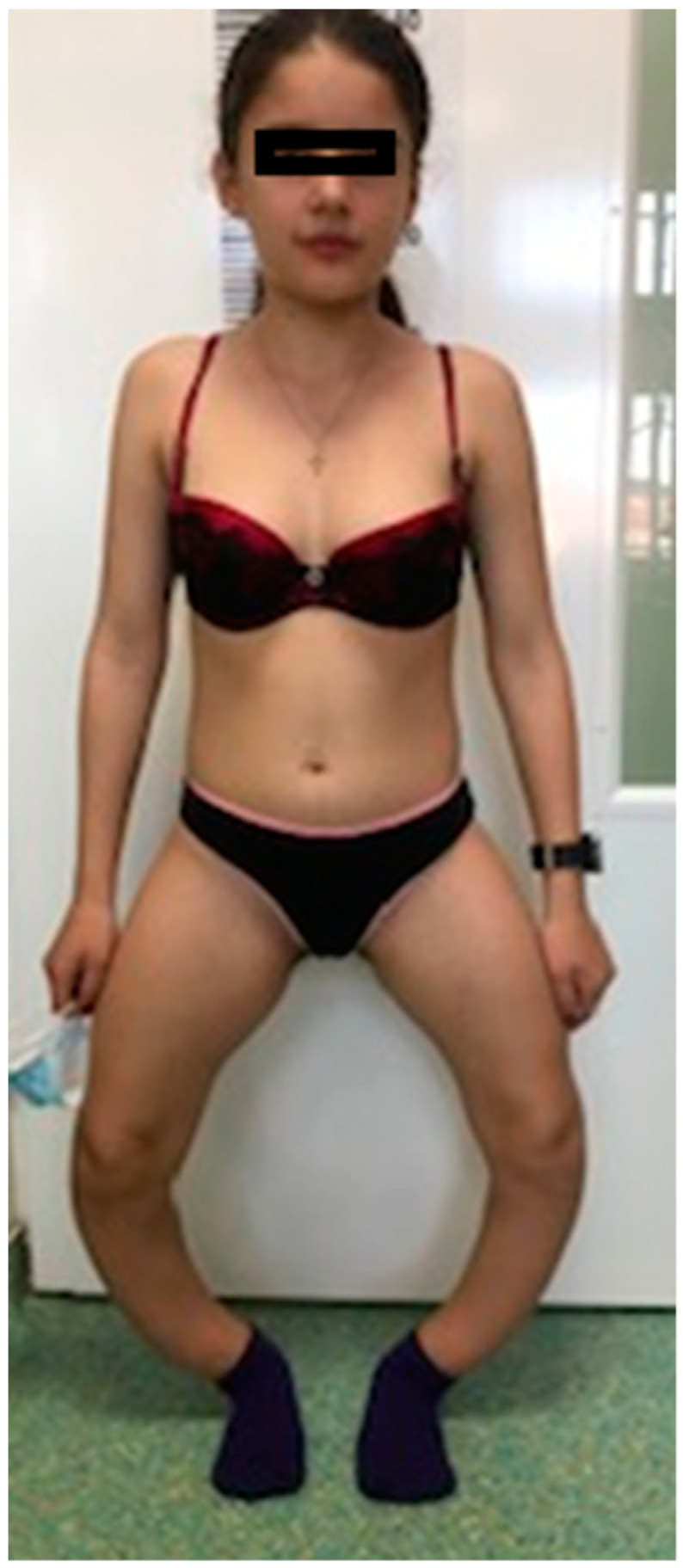
Case #1. Sister at 13½ years old: short stature, genu varum deformation of lower limbs.

**Figure 2 genes-13-01392-f002:**
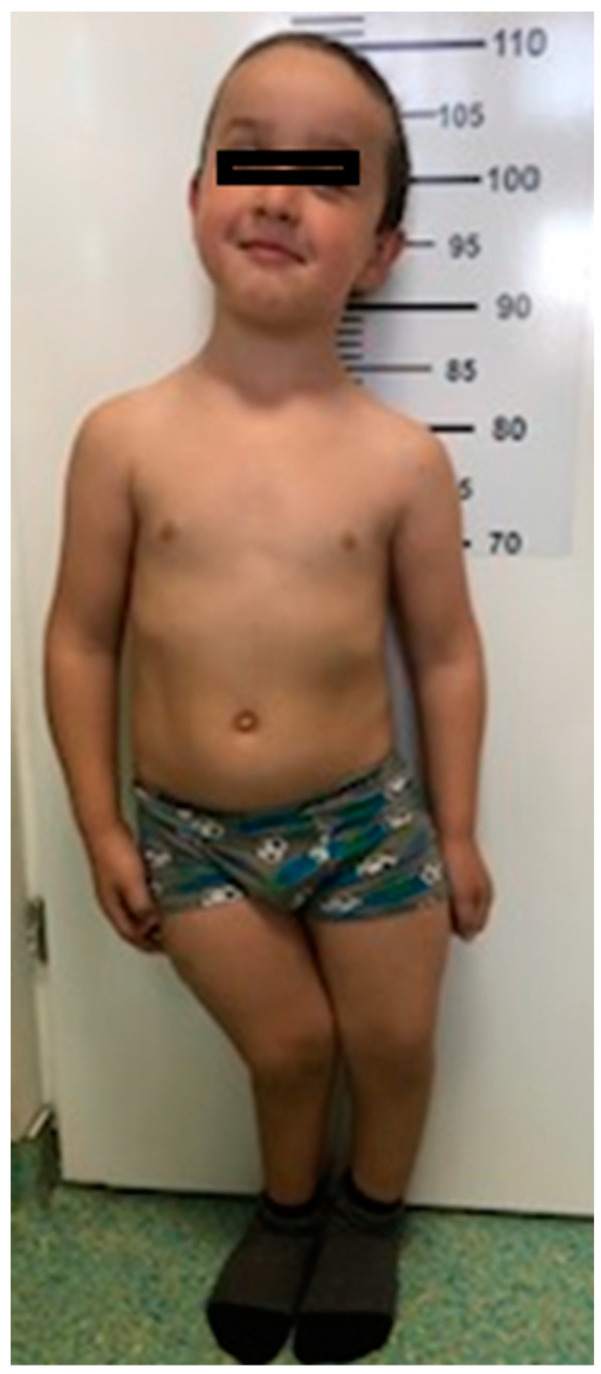
Case #2. Brother at 7 years old: short stature, macrocephaly, genu valgum deformation of lower limbs, knees touching each other while ankles remain spaced apart.

**Figure 3 genes-13-01392-f003:**
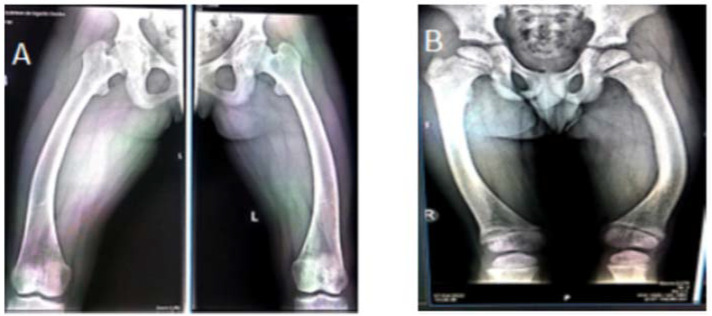
Femoral radiography (April 2022): femoral scoliosis, (**A**) case #1, (**B**) case #2.

**Figure 4 genes-13-01392-f004:**
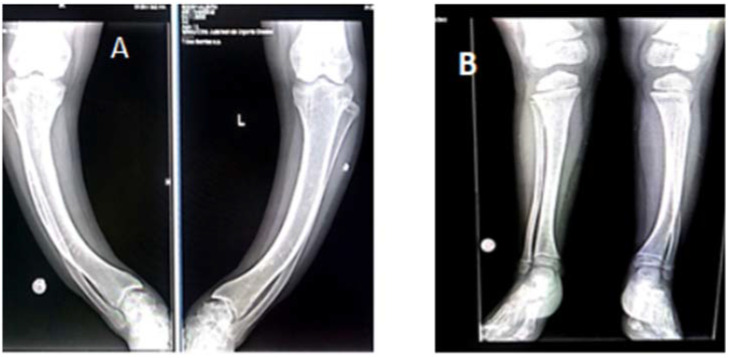
Tibial and fibular radiography (April 2022). (**A**): Case #1: tibial scoliostosis, mild bilateral fibular deformation, closed growth plates. (**B**): Case #2: left tibia: scoliostosis in varum, right tibia: scoliostosis invalgum.

**Figure 5 genes-13-01392-f005:**
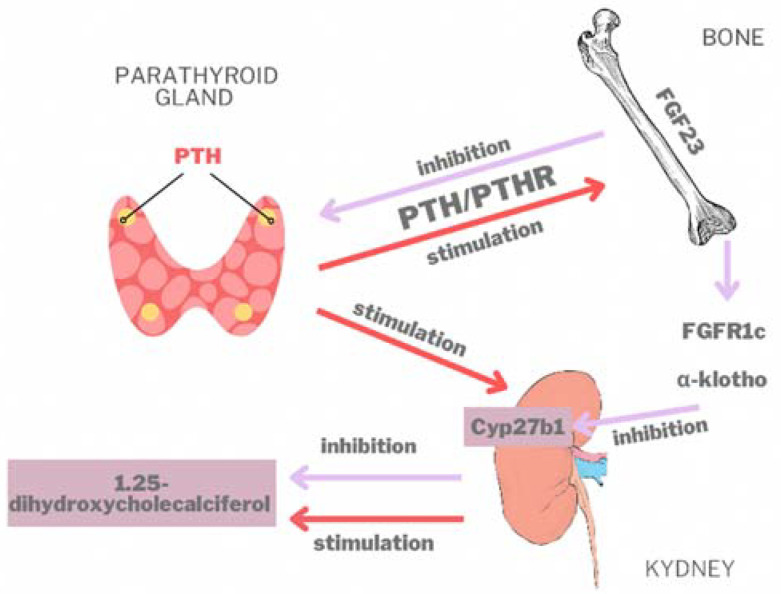
The role of FGF23 in circulation; PTH-parathormone; PTHR-receptor of parathormone.

**Table 1 genes-13-01392-t001:** Morphological evaluation of the patients at the initiation of treatment with Burosumab.

	Case #1 Sister	Case #2 Brother
Age at diagnosis	13½ years	7 years
Age of onset	3 years	2 years
Weight	39 kg	19 kg
Stature	131 cm (−3DS)	107 cm (−3DS)
Craniofacial dysmorphism		macrocephaly: head circumference 53 cm frontal bossing flattened facies bilateral epicanthal folds, high arched palate
Chest deformity	pectus excavatum, thoracolumbar scoliosis, Harrison’s groove and rachitic rosary	pectus excavatum, Harrison’s groove and rachitic rosary
Musculoskeletal abnormalities	thickened wrists and ankles, severe bilateral bowing of lower limbs (genu varum)	thickened wrists and ankles, genu valgum deformity (knees touching each other while the ankles remain spaced apart)
Walk	waddling gait	impaired
Walking fatigue	++++++	+++

**Table 2 genes-13-01392-t002:** Laboratory investigations.

Investigations (Reference Values)	Case#1		Case#2	
	Initiation of Treatment with Burosumab	1 Year after Starting Treatment with Burosumab	Initiation of Treatment with Burosumab	1 Year after Starting Treatment with Burosumab
Phosphorus (2.4–4.4 mg/dL)	1.85	2.6	2.41	3.0
Alcaline phosphatase (3–10 years: 130–260 U/L, 10–14 years 130–340 U/L)	488	159	788	379
Total Calcium (8.8–10.8 mg/dL)	10.2	9.5	9.9	9.40
FGF23 (26–110 kRU/l)	201		215	
Parathormone (PTH) (12–65 pg/mL)	24	63	63.1	94
1,25(HO)_2_ dehydrogenase (25–86 pg/mL)	59.5	83.7	62.7	89.10
Glomerular filtration rate (over 90 mL/min)	122.3	142.7	134.5	149

**Table 3 genes-13-01392-t003:** Radiological investigations.

	Initiation of Treatment with Burosumab	1 Year after Starting Treatment with Burosumab
**Case# 1**	Chest: discrete dextroconvex dorsolumbar scoliosis Bilateral femur: bilateral femural scoliostosis, bilateral enlargement of the distal metaphysis and epiphysis. Bilateral knee, leg, ankle: major bilateral tibial scoliostosis; mild bilateral fibular deformation. Bilateral enlargement of the proximal and distal tibial epiphysis and metaphysis Bone demineralization of the radiographed skeleton, with fine opaque lines Bone age corresponds to chronological age	Bilateral femural scoliostosis (in varum); Marked bilateral tibial scoliostosis; mild bilateral fibular deformation. Closed growth plates. Bone age corresponds to chronological age
**Case # 2**	Chest: Marked bilateral widening of the anterior ends of the ribs. Costal rosaries. Femur: Deformed, curved femoral diaphysis, widened distal metaphysis and irregular contour at growth plates level. Deformed, curved fibular and tibial diaphysis with widened metaphysis and irregular contours Bone age corresponds to chronological age	Bilateral femoral scoliostosis (in varum); Widened femoral metaphysis In varum scoliostosis of the left tibia, in valgum scoliostosis of the right tibia. Bone age corresponds to chronological age

**Table 4 genes-13-01392-t004:** The c.565C > T (p.Gln189Ter) mutation in literature.

Author	Title	References
Yamazaki Y et al.	Increased circulatory level of biologically active full-length FGF-23 in patients with hypophosphatemic rickets/osteomalacia	[34]
Zhang C et al.	Clinical and genetic analysis in a large Chinese cohort of patients with X-linked hypophosphatemia.	[35]
Morey M et al.	Genetic diagnosis of X-linked dominant Hypophosphatemic Rickets in a cohort study: tubular reabsorption of phosphate and 1,25(OH)2D serum levels are associated with PHEX mutation type	[36]
Vila-Pérez D et al.	Four Cases of X-Linked Hypophosphatemic Rickets, Clinical Description and Genetic Testing	[37]

**Table 5 genes-13-01392-t005:** Open-label studies and multiple case reports about the effects of the Burosumab treatment.

Study and Patients	Results	Observations	References
Open-label phase 2 trial No. of patients: 52 Age: 1–12 Tracking period: 64 weeks	increased serum phosphate levels low-amplitudine bone pain increased capacity for effort	Positive effects of Burosumab treatment.	[73]
Open-label phase 3 trial at 16 clinical sites No. of patients: 61: 29 received Burosumab, 32 conventional therapies Age: 1–12 Tracking period: 64 weeks	improvement in rickets and long bone deformities amelioration in linear growth increased serum phosphate levels decreased renal phosphate loss	Burosumab offers a promising new treatment approach for children with XLH in comparison with conventional therapy	[75]
Open-label phase 2 trial at three hospitals in the US No. of patients: 13 Age: 1–4 Tracking period: 64 weeks	increased serum phosphate levels decreased severity of rickets prevention of growth decline	Burosumab had a favorable safety profile	[77]

## Data Availability

Not applicable.

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
