# Peer review of "Effects of Burosumab Treatment on Two Siblings with X-Linked Hypophosphatemia. Case Report and Literature Review"

_genes, 2022, doi:10.3390/genes13081392_

Round 1

Reviewer 1 Report

Comment to Authors:

The manuscript entitled “Effects of Burosumab treatment on two siblings with X-linked 2 hypophosphatemia. Case report and literature review “ reports about the treatment of two siblings with Burosumab.

Comments:

-Please add a “:” in line 79,

-What means “cu” in line 113?

-Figure 5: I would delete this figure in discussion section

-Dental X-ray: It would be very interresting, if available, to see Dental X-rays of the patients!

-Discussion/Introduction section: please add some information about oral health related quality of life in patients with XLH

Author Response

The manuscript entitled “Effects of Burosumab treatment on two siblings with X-linked 2 hypophosphatemia. Case report and literature review “ reports about the treatment of two siblings with Burosumab.

Comments:

-Please add a “:” in line 79,

Thank you, we corrected.

-What means “cu” in line 113?

Thank you, it is an error, cu = rom word means with

-Figure 5: I would delete this figure in discussion section

We changed the picture 5 with a corrected one.

-Dental X-ray: It would be very interresting, if available, to see Dental X-rays of the patients!

Thank you, unfortunately we don’t have any X-ray for these patients, they are healthy from this point if view.

-Discussion/Introduction section: please add some information about oral health related quality of life in patients with XLH.

We add at  r 247-252.

Reviewer 2 Report

It is inaccurate to state in the abstract that "It is caused by mutations present in the Phosphate Regu-
lating Endopeptidase Homolog X-Linked (PHEX) gene responsible for the degradation of the bone-derived hormone fibroblast growth factor 23 (FGF23) into inactive fragments."

The mechanism by which the PHEX gene product affects circulating FGF23 levels is unknown and is may be an indirect mechanism- the PHEX gene product, although an endopeptidase, has been shown to not impact FGF23 as a substrate for PHEX enzyme activity. The mechanism is currently unclear. This is more clearly stated in the body of the paper.

Grammar correction "The clinical picture includes hypophosphataemia, disharmonic dwarfism, bone deformities (curvature of the lower limbs), osteoalgia, dental abscesses, osteomalacia, and enthesopathies."  The enthesopathy and osteoarthritis are equally reported in the literature and represents progression of the disease into adulthoood.

Patients should be identified by sex, rather than gender.

Chemistries, inheritance patterns, physical presentation and genetic analysis confirm the diagnosis of XLH in the 2 subjects and are comprehensive and very well presented in this paper.  The review of the current literature and therapies for XLH are well-written and cited, although the level of detail of cited literature is unusual for a case study.   The editors can comment on length if there are word constraints.

Spelling: Besides phosphate and 1,25(OH) 2 D3, other regulatory factors such as calcium, leptin, estrogen, glucocorticoids, iron metabolism and bone mineralization are involved in FGF23 expression

In general, there are no novel findings in this paper regarding etiology, patient presentation, inheritance patterns, treatment outcomes with Burosumab- however, given that degree of lapses in diagnosis and recognition of the disorder in Romania, this paper may serve as a catalyst for better dissemination of this information to the affected population.

Author Response

It is inaccurate to state in the abstract that "It is caused by mutations present in the Phosphate Regu-
lating Endopeptidase Homolog X-Linked (PHEX) gene responsible for the degradation of the bone-derived hormone fibroblast growth factor 23 (FGF23) into inactive fragments."

The mechanism by which the PHEX gene product affects circulating FGF23 levels is unknown and is may be an indirect mechanism- the PHEX gene product, although an endopeptidase, has been shown to not impact FGF23 as a substrate for PHEX enzyme activity. The mechanism is currently unclear. This is more clearly stated in the body of the paper.

Thank you, we corrected.

Grammar correction "The clinical picture includes hypophosphataemia, disharmonic dwarfism, bone deformities (curvature of the lower limbs), osteoalgia, dental abscesses, osteomalacia, and enthesopathies."  The enthesopathy and osteoarthritis are equally reported in the literature and represents progression of the disease into adulthoood.

Patients should be identified by sex, rather than gender.

Chemistries, inheritance patterns, physical presentation and genetic analysis confirm the diagnosis of XLH in the 2 subjects and are comprehensive and very well presented in this paper.  The review of the current literature and therapies for XLH are well-written and cited, although the level of detail of cited literature is unusual for a case study.   The editors can comment on length if there are word constraints.

Spelling: Besides phosphate and 1,25(OH) 2 D3, other regulatory factors such as calcium, leptin, estrogen, glucocorticoids, iron metabolism and bone mineralization are involved in FGF23 expression.

Thank you, we corrected.

In general, there are no novel findings in this paper regarding etiology, patient presentation, inheritance patterns, treatment outcomes with Burosumab- however, given that degree of lapses in diagnosis and recognition of the disorder in Romania, this paper may serve as a catalyst for better dissemination of this information to the affected population.